# Rice Peroxygenase-9 Negatively Regulates Production of Reactive Oxygen Species and Increases Cellular Resistance to Abiotic Stress

**DOI:** 10.3390/ijms26146918

**Published:** 2025-07-18

**Authors:** Anh Duc Tran, Kyoungwon Cho, Manh An Vu, Jeong-Il Kim, Hanh Thi Thuy Nguyen, Oksoo Han

**Affiliations:** 1Graduate School and Kumho Life Science Laboratory, Department of Integrative Food, Bioscience and Biotechnology, College of Agriculture and Life Sciences, Chonnam National University, Gwangju 61186, Republic of Korea; anhduchytq1997@gmail.com (A.D.T.); kw.cho253@gmail.com (K.C.); manhanhd98@gmail.com (M.A.V.); kimji@jnu.ac.kr (J.-I.K.); 2Faculty of Biotechnology, Vietnam National University of Agriculture, Hanoi 12406, Vietnam; ntthanh.bio@gmail.com; 3Department of Molecular Biotechnology, Chonnam National University, Gwangju 61186, Republic of Korea

**Keywords:** abiotic stress, antioxidant enzymes, enzyme mechanism, epoxy fatty acid, oxylipin, plant lipid, plant defense, reactive oxygen species, rice peroxygenase, unsaturated fatty acid

## Abstract

Caleosin/peroxygenases (CLO/PXGs) play critical functional roles during plant development, oxylipin metabolism, and the response to abiotic/biotic stressors and environmental toxins. In *Oryza sativa*, peroxygenase-9 (OsPXG9) catabolizes intermediates in oxylipin biosynthesis produced by lipoxygenase-9 (9-LOX) and scavenges HOOH and CuOOH by transferring oxygen to hydroxy fatty acids (HFAs) but not to the free fatty acids. The resulting epoxide derivatives of HFAs are then enzymatically or non-enzymatically hydrolyzed into the corresponding trihydroxy derivatives. Results presented here demonstrate OsPXG9′s specificity for catabolizing products of the 9-LOX (and not for the 13-LOX) pathway of oxylipin biosynthesis. Overexpression of *OsPXG9* reduces ROS (reactive oxygen species) abundance and reduces drought- and salt-stress-induced apoptotic cell death. The high expression level of *OsPXG9* also stimulates drought- and salt-induced but not basal expression of antioxidant enzymes/pathways in plants, thereby increasing cellular resistance to drought. These results suggest that OsPXG9 decreases ROS abundance and is essential to increase resilience in rice plants exposed to exogenous or endogenous abiotic stress.

## 1. Introduction

Caleosin/peroxygenases (CLO/PXG, EC 1.11.2.3) are a conserved family of proteins [1] initially detected at least in part due to their ubiquitous presence and structural role in plant lipid bodies [2,3,4], while later studies focused on CLO/PXG’s intrinsic peroxygenase activity. CLO/PXGs are referred to interchangeably as caleosin (CLO) or peroxygenase (PXG) enzymes [1]. Enzymatic functions of CLO/PXG proteins include the hydroxylation of aromatic compounds [5], sulfoxidation of methiocarb [6], and epoxidation of polyunsaturated fatty acids (PUFAs) [7], essential heme ligand-dependent steps during oxylipin biosynthesis and metabolism [8,9]. Unlike cytochrome P450, whose single conserved cysteine residue acts as a fifth axial heme ligand, CLO/PXG proteins harbor a highly conserved heme-binding domain with two invariant histidine residues acting as fifth (proximal) and sixth (distal) ligands of the heme-coordinated ferrous ion (Fe(II)) [9]. Mechanistic studies reveal that PXG catalyzes a two-step reaction, as follows: step 1 involves heterolytic cleavage of the hydroperoxide O-O bond results to produce the corresponding alcohol and a transient ferryl-oxo intermediate that is released at the PXG active site [10]; in the second step, PXG catalyzes intramolecular or intermolecular epoxidation of PUFA substrates, producing epoxy fatty acids, as described previously [11].

Plant oxylipins are generated by endogenous enzymatic and non-enzymatic biosynthetic pathways from PUFA substrates [12,13,14]. The biology and molecular functions of oxylipins in plants have been studied extensively, revealing that oxylipins are essential for normal plant development and plant survival in the presence of exogenous and endogenous stressors. The first steps in the enzymatic biosynthesis of oxylipins involve sequential 9- or 13-lipoxygenase (9-LOX or 13-LOX) followed by 9- or 13-peroxygenase (9-PXG or 13-PXG), respectively, or alternative enzymes/pathways downstream of 9-LOX or 13-LOX [15,16]. Peroxygenase enzymes from different plants have variable capacity/substrate specificities for LOX reaction products; for example, 9-LOX and 13-LOX-specific downstream pathways have been identified and characterized [8,10,17,18,19]. The products include mono-, di-, and poly-epoxy FAs, which can subsequently be converted into hydroxy FAs [10,20,21].

As mentioned above, CLO/PXG enzymes play pivotal roles in plant development and increase plant survival and resilience in the presence of exogenous and endogenous stressors. In rapeseed and *Arabidopsis*, CLO/PXG enzymes support and promote trafficking and metabolism of intracellular lipids [22]. CLO/PXGs also play roles in seed germination, tissue differentiation in seedlings, leaf senescence, pollen maturation, and seed filling [23,24,25,26,27,28,29]. In *Arabidopsis*, *AtPXG3/RD20* is upregulated by exposure to high salt or drought conditions, and overexpression of *AtPXG3/RD20* increased the rate of transpiration and reduced drought tolerance [30]. In date palm, two tissue-specific CLO/PXG isoforms were reported to facilitate detoxification in plants exposed to dioxins. There is also evidence that CLO/PXG enzymes support detoxification after exposure to biotic stress, such as xenobiotics, antifungal compounds, and cutin monomers [31,32].

Our previous study shows that OsPXG9 catalyzes LOX-dependent regiospecific epoxidation of lipid peroxides, identifies and reports kinetic parameters for LOX-generated products, and characterizes expression of *OsPXG9* in the presence and absence of abiotic or biotic stressors [19]. Here, we expand our previous study and characterize the substrate specificity and quantify product distribution and abundance with more substrates of OsPXG9. OsPXG9′s role in preventing oxidative stress/oxidative damage in plants exposed to abiotic stressors is also discussed. Thus, we expect that the results of the present study will elucidate the molecular mechanism(s) by which OsPXG9 improves plant resilience/survival, as well as its critical role(s) during oxylipin biosynthesis. In particular, this study provides new insights into the catalytic preferences of OsPXG9 for different substrate types and its physiological function in response to abiotic stress, thereby complementing and extending our previous findings.

## 2. Results

### 2.1. Kinetic Analysis of OsPXG9-Catalyzed Inter- and Intra-Molecular Oxygen Transfer

The catalytic activity of OsPXG9 in both intramolecular and intermolecular oxygen transfer reactions has been characterized previously [19]. Using hydroperoxy fatty acids, such as 9(*S*)-HPOD(T)E, as substrates, OsPXG9 was shown to favor intramolecular oxygen transfer over intermolecular reactions involving cumene hydroperoxide (CuOOH) as an oxygen donor and aniline as an acceptor. This preference was further supported by UV spectroscopic analysis, which revealed a progressive decline in the levels of oxygen acceptors, specifically L(n)A derivatives (9/13(*S*)-HOD(T)E), as well as the oxygen donors HOOH and CuOOH (Appendix A). Although UV absorbance at 234 nm remained constant when 13(*S*)-HOTE was the oxygen acceptor (Appendix A), it is possible that OsPXG9 epoxidizes the double bond at C15 of 13(*S*)-HOTE without altering the conjugated diene. The kinetic parameters of the intermolecular oxygen transfer reactions are shown in Appendix A and summarized in Table 1.

The current results demonstrate that the OsPXG9 k_cat_ and k_cat_/K_m_ for CuOOH were 4 and 5 times higher, respectively, than for HOOH in reactions with 9(*S*)-HODE. Similarly, k_cat_ and k_cat_/K_m_ for CuOOH in reaction with 9(*S*)-HOTE were 3 and 11 times higher, respectively, than for HOOH. This supports the conclusion that OsPXG9 preferentially uses CuOOH as an oxygen donor in 9-LOX pathway reactions. When 13(*S*)-HODE was the oxygen acceptor, A_234 nm_ was stable during reactions with CuOOH, the same as observed previously during intramolecular transfer using 13-HPODE. However, 13(*S*)-HODE abundance decreased slowly in reactions with OsPXG9 and HOOH. The k_cat_/K_m_ of OsPXG9 for HOOH in reactions with 13(*S*)-HODE were 18 and 14 times lower than for 9(*S*)-HODE and 9(*S*)-HOTE, respectively (Table 1). Thus, the results support the conclusion that OsPXG9 preferentially uses CuOOH (vs. HOOH) as an oxygen donor in this pathway.

Similar analysis for oxygen acceptors and OsPXG9 demonstrated that k_cat_ and k_cat_/K_m_ were significantly lower for hydroxy fatty acids than for hydroperoxy fatty acids. Earlier studies of the 9-LOX pathway showed a 4-fold higher k_cat_/K_m_ for OsPXG9 and 9(*S*)-HPOTE than with 9(*S*)-HPODE [19]. However, in this current study, k_cat_/K_m_ for 9(*S*)-HODE and 9(*S*)-HOTE were similar in the reactions with HOOH and only slightly higher for 9(*S*)-HODE using CuOOH as the oxygen donor. For 13-LOX pathway substrates, k_cat_ for 13(*S*)-HODE was higher than for 9(*S*)-HODE and 9(*S*)-HOTE, while k_cat_/K_m_ of the 13-LOX pathway was lower than in the 9-LOX one. Overall, OsPXG9 preferentially uses hydroxy fatty acids in the 9-LOX pathway, confirming our previous findings [19]. In contrast, kinetic parameters of OsPXG9 were similar in reactions with 9(*S*)-HODE and 9(*S*)-HOTE (Table 1).

### 2.2. Products of Intermolecular Oxygen Transfer Using Newly Validated Substrates

#### 2.2.1. Analysis of OsPXG9 Primary Reaction Products by TLC

Reaction products were extracted and analyzed by TLC, as described in the 4. Materials and Methods Section. TLC data revealed that neither HOOH nor CuOOH supported OsPXG catalysis in reactions with OA, LA, or LnA (Appendix A). However, OsPXG9 accepted HOOH as an oxygen donor in reactions with 9(*S*)-HODE, 9(*S*)-HOTE, 13(*S*)-HODE, and 13(*S*)-HOTE, generating 3, 3, 3, and 1 product, respectively. Using CuOOH as an oxygen donor, OsPXG9 converted 9(*S*)-HODE, 9(*S*)-HOTE, and 13(*S*)-HOTE to 2, 2, and 1 product; in contrast, no product was detected by TLC in reactions with 13(*S*)-HODE. Reaction products were analyzed by LC-MS/MS, as described below.

#### 2.2.2. Analysis of OsPXG9 Reaction Products in the 9-PXG Pathway

Products of the OsPXG9 reaction were analyzed by LC-MS/MS using negative ion mode. For the reactions using HOOH as an oxygen donor, the substrate 9(*S*)-HODE was detected at 10.52 min with m/z 295.2 (peak 1 in Figure 1A), and the mass spectrum was confirmed by comparison with the mass bank library (Appendix A). Products from 9(*S*)-HODE were detected as m/z 329.3 (peaks 2 and 3) at 8.40 and 8.57 min, 311.3 (peak 4) at 9.33, and 311.3 (peak 5) at 9.50 min (Figure 1B), and identified as 9(*S*)-9,12,13-trihydroxy-10-octadecenoic acid (9(*S*)-9,12,13-THOE), 9(*S*)-10,11-epoxy-9-hydroxy octadecenoic acid (9(*S*)-10,11-EHOE), and 9(*S*)-12,13-epoxy-9-hydroxy octadecenoic acid (9(*S*)-12,13-EHOE), respectively (Appendix A and Table 2).

For 9(*S*)-HOTE, the substrate was detected at 10.19 min with m/z 293.2 (peak 6 in Figure 1C), and the identity was confirmed (Appendix A). Products generated from 9(*S*)-HOTE were detected as m/z 327.2 (peaks 7 and 8) at 8.13 and 8.18 min, 309.2 (peak 9) at 8.94 min, and 309.2 (peak 10) at 9.18 min, and confirmed as 9(*S*)-9,12,13-trihydroxy-10,15-octadecadienoic acid (9(*S*)-9,12,13-THODE), 9(*S*)-10,11-epoxy-9-hydroxy-12-octadecadienoic acid (9(*S*)-10,11-EHODE), and 9(*S*)-12,13-epoxy-9-hydroxy-10-octadecadienoic acid (9(*S*)-12,13-EHODE), respectively (Appendix A and Table 2). Peak 8 was isolated by silica column chromatography and its structure was further characterized by proton NMR. As shown in the Appendix A, analysis of chemical shifts and coupling constants clearly indicated peak 8 as the single diastereomer of 9(*S*)-9,12,13-THODE.

In reactions using CuOOH as an oxygen donor, a similar distribution of substrates and products was detected (Figure 1E–H), and reaction products were also confirmed by MS/MS analysis (Appendix A and Table 2).

#### 2.2.3. Analysis of OsPXG9 Reaction Products in the 13-PXG Pathway

For 13-PXG pathway reactions, the LC chromatogram detected the substrate (13(*S*)-HODE) at 10.49 min with m/z 295.2 (peak 11 in Figure 2A,E), and the substrate’s identity was confirmed (Appendix A). For product distribution, in the case of the reaction between 13(*S*)-HODE and HOOH, three peaks (peaks 12, 13, and 14) were detected at 8.49, 8.54, and 9.35, respectively (Figure 2B). Peaks 12 and 13 with the m/z of 329.2 were identified as 13(*S*)-9,10,13-trihydroxy-11-octadecenoic acid (13(*S*)-9,10,13-THOE), while peak 14 with the m/z of 311.2 was identified as 13(*S*)-9,10-epoxy-13-hydroxy octadecenoic acid (13(*S*)-9,10-EHOE) (Appendix A and Table 2). However, no product formation was observed in the reaction between 13(*S*)-HODE and CuOOH (Figure 2F).

For 13(*S*)-HOTE, this substrate was detected as m/z 293.2 at 10.20 min (peak 15 in Figure 2C,G) and confirmed by MS/MS (Appendix A). Regardless of using HOOH or CuOOH, only one peak of the product was detected as m/z 309.2 at 9.29 min (peak 16 in Figure 2D,H) and determined as 13(*S*)-15,16-epoxy-13-hydroxy-9,11-octadecadienoic acid (13(*S*)-15,16-EHODE) (Appendix A and Table 2).

#### 2.2.4. Relative Abundance of Products from 9- and 13-PXG Pathways Catalyzed by OsPXG9

In previous work, the relative abundance of products formed from 9(*S*)-HPODE, 9(*S*)-HPOTE, and 13(*S*)-HPOTE was determined based on LC chromatogram peak areas [19]. In this study, we extend that analysis by quantifying the relative product abundance from reactions involving 9(*S*)-HODE, 9(*S*)-HOTE, 13(*S*)-HODE, and 13(*S*)-HOTE with either HOOH or CuOOH, and comparing these results with those of the previously studied hydroperoxy substrates (Table 3). Overall, the 9-PXG pathway showed the highest relative rate constants (K_rel_) when hydroperoxy fatty acids were used as substrates, followed by reactions involving hydroxy fatty acids with HOOH, and lastly, the reactions between hydroxy fatty acid (9(*S*)-HODE and 9(*S*)-HOTE) and CuOOH had the lowest K_rel_ values. Specifically, for linoleic acid (LA)-derived compounds, the K_rel_ of 9(*S*)-HPODE was approximately 3-fold and 10-fold higher than that of reactions using 9(*S*)-HODE with HOOH or CuOOH, respectively. Similarly, for linolenic acid (LnA)-derived substrates, the K_rel_ of 9(*S*)-HPOTE was about 9 times and 13 times greater than that of reactions involving 9(*S*)-HOTE with HOOH or CuOOH, respectively. In the 13-PXG pathway, OsPXG9 selectively utilized 13(*S*)-HODE as a substrate only when HOOH was the oxygen donor, but its K_rel_ was significantly lower by 7-, 4-, and 10-fold compared to 9(*S*)-HODE, 9(*S*)-HOTE, and 13(*S*)-HOTE, respectively. Interestingly, in the case of 13(*S*)-HPOTE and 13(*S*)-HOTE, the reaction using 13(*S*)-HOTE with HOOH exhibited a K_rel_ of approximately 30 times greater than that observed with either 13(*S*)-HPOTE or 13(*S*)-HOTE when CuOOH was used (Table 3).

### 2.3. Overexpression of OsPXG9 Reduces ROS Abundance and Lowers the Frequency of Apoptotic Cell Death After Exposure to Abiotic Stress

OsPXG9 is highly upregulated in rice seedlings exposed to drought and/or salt stress [19], and such ROS bursts can significantly reduce cell viability and induce cell death [35,36]. Here, we investigated the biological role of OsPXG9 by overexpressing the enzyme in rice (*Oryza sativa L. Japonica* cv. ‘Ilmi’) via Agrobacterium tumefaciens-mediated transformation. Thus, transgenic overexpression lines were generated and characterized, as shown in Appendix A. Two transgenic overexpressing plants were identified for further study among approximately 2000 Agrobacterium tumefaciens-transformed calli. The OsPXG9 copy number was quantified by qRT-PCR in WT and two T_0_ transgenic overexpression lines of interest. The results showed 1.12 and 2.45 gene copies in ox1 and ox2 transgenic plants, respectively (Appendix A), suggesting that the ox2 line carries three genomic transgenes expressing OsPXG9, while ox1 plants lack a detectable transgene. OsPXG9 was 1.3- and 3.3-fold higher in ox1 and ox2 than in WT, respectively (Appendix A), consistent with the OsPXG9 copy number detected in the transgenic lines. For the T_1_ generation, two plants (ox2-1 and ox2-2) were successfully grown from seeds of the ox2 transgenic plant. Expression of OsPXG9 in ox2-1 and ox2-2 plants was 106.5- and 2.6-fold higher than in WT, respectively (Appendix A). Seeds collected from ox2-1 and ox2-2 plants, denoted ox2-1 and ox2-2, respectively, were used in the following experiments. The ox2-1 and ox2-2 plants were significantly more tolerant to salt stress (200 mM NaCl) than WT (Appendix A).

DAB histochemical staining revealed reduced ROS accumulation in the ox2-1 and ox2-2 mutants compared to wild-type plants following drought or salt stress. Similarly, less abundant and less intense NBT staining was detected in ox2-1/ox2-2 plant cells than in WT, likely reflecting less abundant ROS in the leaves of transgenic rice. These results suggest that the abundance of HOOH and O_2_^•−^ is lower in transgenic plants overexpressing OsPXG9 than in WT (Figure 3A,B). Consistent with this, fewer apoptotic cells were detected with Evan’s blue stain in the transgenic plants than in WT controls after exposure to abiotic stress (Figure 3C). Finally, the abundance of malondialdehyde (MDA), a marker for lipid peroxidation, was also statically lower after abiotic stress in both the OsPXG9-overexpressing plants than in WT. Notably, MDA abundance was 2-fold lower in drought- or salt-stressed ox2-1 plants than in drought- or salt-stressed WT plants. MDA was also lower in stressed ox2-2 plants, but to a lesser extent than ox2-1 plants (Figure 3D).

### 2.4. Overexpression of OsPXG9 Correlates with Reduced Expression of Antioxidant Pathways in Unstressed Plants

High levels of ROS in response to exogenous or endogenous stress typically cause peroxidation of membrane lipids and other types of cellular damage, leading to stress-induced cell death. Thus, ROS abundance is tightly and cooperatively regulated to maintain cellular homeostasis. To better understand co-regulation of OsPXG9 and ROS, we investigated expression of antioxidant-related enzymes: ascorbate peroxidase (APX), catalase (CAT), superoxide dismutase (SOD), and glutathione reductase (GR), and their corresponding genes (*OsAPX1, OsCAT-A, OsSOD1*, and *OsGR*), by qRT-PCR [37]. In unstressed plants, expression of these genes was approximately 4-fold lower in ox2-1 plants (statistically significant values of 5.1, 4.2, 3.8, and 5.5 times for *OsAPX1*, *OsCAT-A*, *OsSOD1*, and *OsGR*, respectively) than in WT control plants, while in ox2-2 plants, expression of OsCAT-A and OsGR was suppressed (statistically significant values of 7.0 and 1.9 times, respectively). In contrast, after exposure to drought stress, ox2-1 plants and WT plants expressed similar levels of these four genes, while in ox2-2 plants, expression of OsSOD1 did not change, but expression of OsAPX1, OsCAT-A, and OsGR was upregulated 2- to 3-fold. In response to salt stress, expression of OsAPX1, OsSOD1, and OsGR was 2- to 3-fold higher in transgenic ox2-1 plants than in WT controls. Interestingly, after exposure to salt stress, expression of OsCAT-A was somewhat lower in ox2-1 plants than in WT control plants. In salt-stressed ox2-2 plants, expression of OsAPX1, OsCAT-A, and OsGR was similar as in WT control plants, while expression of OsSOD1 was upregulated approximately 6-fold (Figure 4).

## 3. Discussion

### 3.1. In Vitro Studies of OsPXG9-Dependent Hydroperoxide Scavenging Activity

Previous studies have demonstrated that OsPXG9 converts 9-HPOD(T)E and 13-HPOTE into their corresponding epoxy fatty acids, which may subsequently undergo enzymatic or non-enzymatic hydrolysis to form hydroxy fatty acids. While both intra- and inter-molecular oxygen transfer reactions are possible, empirical in vitro data indicate a strong preference—approximately 18:1—for intramolecular epoxidation of 13(*S*)-HPOTE. Nonetheless, OsPXG9 is also capable of catalyzing the epoxidation of aniline in vitro, using CuOOH as an oxygen donor, highlighting the enzyme’s substrate- and context-dependent oxidative activity [19]. Indeed, other previous studies also show that FAs can also undergo intermolecular oxidation in the presence of HOOH or CuOOH and PXGs from other plant species: specific examples include AsPXG1 [38], AtPXG4 [39], PdPXG2 and PdPXG4 [25], and OsCLO5 [40,41].

To evaluate the validity and relevance of the proposed rationale within the context of this study, OsPXG9 was co-incubated with oleic acid (OA), linoleic acid (LA), or linolenic acid (LnA), along with either HOOH or CuOOH. The reaction mixtures were subsequently analyzed for potential products of OsPXG9-catalyzed oxidation. Results showed that OsPXG9 did not utilize free fatty acids as substrates, regardless of the oxygen donor employed (Appendix A), under the tested experimental conditions. Similar substrate specificity has been previously reported for AtPXG1 [9] and AtPXG3/RD20 [8]. Plant PXGs stratify into two classes based on the location of the hydrophobic domain(s): Class I PXGs, which resemble oleosins, possess a centrally located hydrophobic domain, whereas Class II PXGs feature a hydrophobic domain at the N-terminus [9]. In *Arabidopsis*, there are three Class I PXGs, AtPXG1 through AtPXG3, none of which have activity toward free FAs. In contrast, there are four Class II PXGs in *A thaliana* (AtPXG4 through AtPXG7), all of which exhibit activity toward free FA substrates [39]. Structurally, OsPXG9 is a Class I PXG, conforming to the shared structure of Class I PXGs by having a centrally located hydrophobic domain [19]. The substrate specificity of OsPXG9 is consistent with the characteristic activity of other Class I plant PXGs (as discussed above).

Heme-carrying PXGs and cytochromes (i.e., P450) in plants catalyze intermolecular oxidation of target substrates using HOOH or CuOOH as the oxygen donor in two steps, as follows: first, the hemoprotein is oxidized by the hydroperoxide to form an oxyferryl (Fe^4+^=O) intermediate, and second, the oxidized hemoprotein oxidizes its target substrate [42]. We previously characterized two-step intramolecular OsPXG9-dependent oxidation of its target substrates. To reiterate, in the first step, the hydroperoxy fatty acids (9(*S*)-HPODE, 9(*S*)-HPOTE, and 13(*S*)-HPOTE) entered OsPXG and contributed as the oxygen donor (in the same function with HOOH or CuOOH), resulting in the formation of oxidized OsPXG9 and the respective hydroxy fatty acids (9(*S*)-HODE, 9(*S*)-HOTE, and 13(*S*)-HOTE). In the second step, the double bond in hydroxy fatty acids was then oxidized to form the respective epoxy hydroxy fatty acids (9(*S*)-EHODE, 9(*S*)-EHOTE, and 13(*S*)-EHOTE) as the final products [19]. In the current study, OsPXG9 catalyzed two-step intermolecular epoxidation of target HFA substrates using HOOH or CuOOH as the oxygen donors (Figure 1, Figure 2 and Appendix A).

In the 9-PXG pathway, the reaction between 9(*S*)-HOD(T)E and HOOH or CuOOH produced the same products, with 9(*S*)-9,12,13-THO(D)E as the main products (Figure 1 and Table 2). In this case, trihydroxy FAs are produced by hydrolytic epoxide ring-opening. This reaction is facilitated by anchimeric assistance from 9(*S*)-12,13-EHOE (or by an intramolecular mechanism using target substrate 9(*S*)-HPOD(T)) [19]. Interestingly, a minor amount of 9(*S*)-10,11-EHO(D)E was also detected (Figure 1 and Table 2), suggesting that intermolecular oxygen transfer involving HOOH or CuOOH exhibits lower selectivity for the targeted double bond compared to intramolecular oxygen transfer using 9(*S*)-HPOD(T)E. OsPXG9 had the strongest affinity and the highest catalytic efficiency in reactions with 9(*S*)-HPOTE and 9(*S*)-HPODE [19]. Intermolecular reactions using HOOH or CuOOH as oxygen donors showed lower k_cat_/K_m_ and K_rel_ than the corresponding intramolecular reaction with a hydroperoxy FA. This result suggests that OsPXG9 achieved higher efficiency and had a strong preference for intramolecular vs. intermolecular oxygen transfer under the experimental conditions tested (Table 1 and Table 3 and [19]). OsPXG9 had a 3-fold higher k_cat_/K_m_ and 2-fold higher K_rel_ in reactions with 9(*S*)-HPOTE than in reactions with 9(*S*)-HPODE, although the relative abundance of the corresponding product did not differ as much as expected (Table 1 and Table 3). This could indicate a greater role and effect during the first step of the reaction, namely, during oxidation of OsPXG9 itself, rather than during oxidation of the target HFA substrate. In addition, certain features of the data may be influenced by the small size of HOOH, even though the binding affinity and catalytic efficiency of OsPXG9 was higher in the presence of CuOOH than in the presence of HOOH. Despite this, the relative abundances of the product and K_rel_ were higher when HOOH replaced CuOOH as the oxygen donor in the first step of the pathway (Table 1 and Table 3). Because HOOH’s molecular size and shape are small, it can co-occupy the catalytic pocket along with the target oxygen acceptor/substrate.

In the 13-PXG path, only 13(*S*)-HODE was taken when HOOH was used as the oxygen donor (although with very low k_cat_/K_m_ and K_rel_ compared to other fatty acid derivatives), while the reaction of 13(*S*)-HPODE and the reaction between 13(*S*)-HODE and CuOOH did not produce any detectable product (Table 1 and Figure 2). This can probably be explained as the conformational change due to the formation of the oxoferryl intermediate state after the first step led to the inaccessibility of 13(*S*)-HODE. However, due to the small size of HOOH, the 13(*S*)-HODE and HOOH can enter the OsPXG9 catalytic site concomitantly before the conformational change, and thus the reaction could happen. This phenomenon did not happen when 9(*S*)-HODE, 9(*S*)-HOTE, or 13(*S*)-HOTE worked as the oxygen acceptor (Figure 1 and Figure 2 and Table 2). For 13(*S*)-HPOTE and 13(*S*)-HOTE, the epoxidation always happened at the double bond between C15 and C16 but not the conjugated double bonds at C9 and C10 or C11 and C12, regardless of the oxygen donor. Therefore, it is worthwhile to assume that in the second step of the oxidation, all the hydroxy fatty acids enter the catalytic cavity of OsPXG9 from the methyl end (ɷ-end), and the –OH group might be the obstacle that prevents the full entry of the oxygen acceptors, resulting in that once the conformation of OsPXG9 is changed, the oxidation can only happen to the double bond(s) located between the C–OH position and the methyl end of the oxygen acceptor.

The kinetic parameters of OsPXG9 demonstrated its broad substrate versatility and provided key insights into its possible physiological functions in planta. OsPXG9 showed the highest catalytic efficiency (k_cat_/K_m_) with 9(*S*)-HPOTE (25.5 s^−1^ μM^−1^) and 9(*S*)-HPODE (7.0 s^−1^ μM^−1^) [19], indicating that hydroperoxy fatty acids are likely its most favorable natural substrates. These lipid hydroperoxides are commonly generated during membrane lipid peroxidation under abiotic stress, supporting the idea that OsPXG9 primarily functions in downstream lipid detoxification and signaling.

While hydrogen peroxide (H_2_O_2_) showed lower catalytic efficiency (Table 1), it still represents a significant substrate for OsPXG9, particularly considering its abundant presence under oxidative stress conditions. More importantly, the reaction using H_2_O_2_ exhibited a high reaction equilibrium constant (Table 3), implying strong thermodynamic favorability. This suggests that OsPXG9 is capable of utilizing HOOH effectively when its cellular levels rise, possibly complementing or interacting with canonical H_2_O_2_-scavenging enzymes like catalase or ascorbate peroxidase, as occurs during drought and salt stress. This ability aligns well with the observed reduction in ROS accumulation and cell death in the overexpression lines under abiotic stress (Figure 3).

Interestingly, CuOOH, a model substrate used in many PXG assays [9], showed relatively low efficiency (Table 1 and Table 3), reinforcing the notion that OsPXG9 is specialized for fatty acid hydroperoxides rather than general peroxides. Additionally, the kinetic parameters varied significantly depending on the oxygen acceptors used. This indicates that OsPXG9 can work with a diverse range of co-substrates, which may reflect its adaptability to various intracellular redox environments.

Taken together, the kinetic profile suggests that OsPXG9 plays a dual role: (i) preferentially targeting lipid hydroperoxides under normal or mild stress conditions, likely participating in signaling or protective oxylipin pathways, and (ii) functioning as a secondary H_2_O_2_ detoxifier under severe oxidative stress, thereby contributing to ROS homeostasis. This mechanistic flexibility likely underpins the physiological protection observed in overexpression lines and aligns with the observed gene expression adjustments in antioxidant pathways (Figure 4).

### 3.2. OsPXG9 Contributes to Regulate ROS Level and Reduces Cell Death upon Drought and Salt Stress

In rice, drought stress and salt stress are commonly known to cause oxidative stress, featured by the accumulation of ROS, causing membrane damage and subsequently leading to cell death [35,43,44]. Previous studies showed that enhancing the ROS scavenging capacity can improve drought stress and salt stress tolerance in plants [45,46,47]. In our previous publication, the expression level of OsPXG9 was highly elevated during drought or salt stress treatment, suggesting that OsPXG9′s main function might be contributing to improving plant performance under abiotic stress conditions [19]. To investigate the biological function of OsPXG9 in vivo, overexpression and CRISPR-Cas9-mediated knockout of *OsPXG9* were performed by *Agrobacterium tumefaciens*-mediated transformation. For overexpression, two T_0_ plants were successfully regenerated from about 2000 transformed calli and were used to generate the T_2_ overexpression lines (ox2-1 and ox2-2 lines, so-called ox2-1 and ox2-2 from now on), which were subsequently used for the next experiments (Appendix A). For CRISPR-Cas9-mediated knockout, regardless of using a single sgRNA or dual sgRNA plasmid, out of 5000 transformed calli transformed with each plasmid, only two transgenic plants were regenerated from calli transformed with the dual sgRNA plasmid. However, both of them were subsequently determined to have no mutation in *OsPXG9′*s DNA sequence. Considering the role of OsPXG9 in ROS regulation in the abiotic stress response (Figure 3 and Figure 4), knocking out of *OsPXG9* possibly results in a decrease in the survival rate of callus during the selection and regeneration stage.

Under drought and salt stress, both ox2-1 and ox2-2 showed decreased accumulation of ROS, including HOOH (Figure 3A), O_2_^•−^ (Figure 3B), and lipid peroxidation (Figure 3D), compared to WT. Similarly, the cell death levels in ox2-1 and ox2-2 were also lower than in WT (Figure 3C). Interestingly, between the overexpression lines, ox2-1 consistently showed better performance in both salt stress and drought stress, most probably due to the higher expression level of *OsPXG9* in ox2-1 compared to ox2-2 (Appendix A). OsPXG9 enzymatically catalyzed the reduction of lipid peroxidation products, including 9(*S*)-HPODE, 9(*S*)-HPOTE, and 13(*S*)-HPOTE [19]. Although OsPXG9 did not show enzymatic activity in reacting HOOH or CuOOH when free fatty acids like OA, LA, or LnA were used as the oxygen acceptors (Appendix A), it readily did so when the oxygen acceptors were 9(*S*)-HODE, 9(*S*)-HOTE, 13(*S*)-HODE, and 13(*S*)-HOTE (Figure 1, Figure 2 and Appendix A). This could suggest that under normal conditions, OsPXG9 does not consume free fatty acid from the membrane and is only activated to work on consuming the products from lipid peroxidation (as the main reaction) or contributing to scavenge other ROS, such as HOOH or O_2_^•−^ (but with a lower rate and substrate preference compared to lipid peroxidation). Better performance in oxidative stress resistance and cell death reduction was also observed in the overexpression lines of *AtPXG3/RD20*, another member of the Class I peroxygenase [8]. In addition, plants from ox2-1 and ox2-2 clearly showed better survival ability in salt stress (Appendix A), implying the connecting function of OsPXG9 in scavenging ROS to cope with abiotic stressors. Overall, OsPXG9, as a member of Class I peroxygenase, probably plays crucial roles in ROS regulation during drought and salt stress.

### 3.3. Overexpression of OsPXG9 Results in Downregulation of Other Antioxidant-Related Enzymes for the Cellular Redox State Balance

The accumulation of ROS in plant cells is commonly known to be related to various environmental stressors, including both biotic and abiotic [48,49]. The accumulation of ROS usually leads to extensive cell damage and cell death, as the excessive ROS causes lipid peroxidation (LPO) by the attack of ^1^O_2_ and OH^•^ to the plasmatic membrane fatty acids (such as linoleic and linolenic acid) in a multistep reaction, leading to the formation of lipid-derived radicals and subsequent disruption of membrane integrity and function [50,51]. To cope with the accumulation of ROS, plants developed a complex array of enzymatic and non-enzymatic detoxification mechanisms. The main enzymes related to antioxidant activity included glutathione peroxidase (GPX), superoxide dismutase (SOD), ascorbate peroxidase (APX), glutathione S-transferase (GST), dehydroascorbate reductase (DHAR), glutathione reductase (GR), peroxide reduction (PRX), mono-hydro ascorbate reductase (MDAR), and catalase (CAT) [52]. In rice, several genes encoding for antioxidant enzymes were reported, including *L-ascorbate peroxidase 1* (*OsAPX1*), *Catalase-A* (*OsCAT-A*), *Superoxide dismutase [Cu-Zn] 1* (*OsSOD1*), and *Glutathione reductase* (*OsGR*) [37,53,54,55,56,57]. Since the overexpression of OsPXG9 results in reducing ROS accumulation and cell death levels under drought and salt stress (Figure 3), the possible relationship between OsPXG9 and other antioxidant enzymes was checked by qRT-PCR. Interestingly, under normal conditions, ox2-1 showed significantly lower expression levels of all four antioxidant genes compared to WT, while in ox2-2, only the expression levels of *OsCAT-A* and *OsGR* showed lower levels. *OsAPX1* and *OsSOD1* showed similar expression levels as those in WT (Figure 4). This phenomenon is possibly associated with the level of *OsPXG9′*s overexpression since *OsPXG9* was much more highly overexpressed in ox2-1 than in ox2-2 (Appendix A). The downregulation of the antioxidant enzymes in *OsPXG9*-ox lines under normal conditions indicated that there are some regulating relationships between OsPXG9 and those antioxidant enzymes, most probably in the functions of regulating the cellular ROS level to maintain the redox state balance (Figure 4). In detail, CAT is directly related to the scavenging of HOOH [58], while GR is also involved in the reduction of HOOH by producing the oxygen acceptor for this reaction [59]. The downregulation of these two genes in both *OsPXG9*-ox lines indicates that there might be a function overlap in regulating the ROS level between *OsPXG9* and *OsCAT-A* as well as *OsGR* to a great extent. The different expression patterns between ox2-1 and ox2-2 suggest a dosage-dependent response in the antioxidant system, potentially reflecting compensatory regulation in response to varying levels of OsPXG9 activity. In ox2-1, where *OsPXG9* is strongly overexpressed, the suppression of multiple antioxidant genes may be due to negative feedback from an already low ROS level, reducing the cellular demand for additional scavenging. In contrast, the milder overexpression in ox2-2 may not lower ROS to the same extent, allowing some antioxidant genes (e.g., *OsSOD1* and *OsAPX1*) to remain actively expressed. This pattern implies that *OsPXG9* may participate in a fine-tuned redox regulation network that adjusts other antioxidant components depending on its own expression level. Furthermore, these changes may not be strictly compensatory or suppressive but could indicate functional crosstalk between OsPXG9 and canonical antioxidant pathways, particularly involving HOOH detoxification. For *OsAPX1* and *OsSOD1*, the downregulations were only observed in the ox2-1 line, indicating that the connections in the function of *OsPXG9* with *OsAPX1* and *OsSOD1* might be weaker than with *OsCAT-A* and *OsGR*. SOD enzymes mainly work on converting the O_2_^•−^ to HOOH [60], providing substrate for HOOH scavenging enzymes; thus, it is reasonable that mildly overexpressing *OsPXG9* does not affect the gene expression of *OsSOD1* in normal conditions. As for APX, these enzymes consume HOOH using ascorbate [61], a different co-substrate from OsPXG9. In addition, apart from ROS scavenging in the cytosol, mitochondria, and peroxisomes, the APX enzymes play a vital role in the recycling of ascorbate and in maintaining the ascorbate–glutathione cycle within chloroplasts [62,63]. Thus, it is possible that a slight HOOH level reduction by overexpression of *OsPXG9* under normal conditions is not sufficient enough to regulate the expression level of *OsAPX1*. In drought conditions, due to the accumulation of ROS, the expression levels of the aforementioned antioxidant enzymes were all restored to similar or slightly upregulated levels compared to those of WT, indicating that the effects of reducing ROS accumulation and cell death levels from Figure 3 could mainly come from the increase in peroxygenase activity by overexpression of *OsPXG9*. Interestingly, under salt conditions, the expression level of *OsCAT-A* remained downregulated in ox2-1 at a similar gene expression level as in non-treated plants (Figure 4). Because CAT enzymes only directly take HOOH as the substrate, they are probably only activated when the cellular HOOH level is increased to a certain threshold [64]. It is possible that at the specific salt stress condition used, due to high peroxygenase activity by overexpression of *OsPXG9* in ox2-1, the accumulated HOOH by SOD was scavenged effectively enough, and the OsCAT-A did not need to be activated (Figure 4). It should be noted that even though HOOH was not the best substrate for OsPXG9, as its kinetic parameters were not as high as the natural substrate (hydroperoxy fatty acids; Table 1), the reactions using HOOH exhibited remarkably high relative equilibrium constant (Table 3), showing a great ability to consume HOOH in the reaction with hydroxy fatty acids. In addition, unlike the reaction using hydroperoxy fatty acids or CuOOH (which does not take 13(*S*)-HPODE or 13(*S*)-HODE as the substrate), the reaction with HOOH showed the peroxygenase activity regardless of the oxygen acceptors (Figure 1 and Figure 2), indicating that for HOOH, OsPXG9 accepts a broader range of oxygen donors and is potentially involved in the HOOH detoxification. As for *OsSOD1*, the gene expression level was upregulated in both ox2-1 and ox2-2 under salt stress, although there has not been direct evidence showing the connection between the expression levels of *SODs* and *PXGs* in plants, and the quick consumption of HOOH due to the overexpression of *OsPXG9* may allow faster conversion of O_2_^•−^ to HOOH by OsSOD1. Thus the upregulation of *OsSOD1* in this situation could allow plants to get rid of the toxic superoxide faster, resulting in less accumulation of both O_2_^•−^, HOOH, and cell death levels in the overexpression lines (Figure 3). Due to the lack of *OsPXG9* knockout lines, this study was limited in its ability to provide direct genetic evidence supporting the regulatory role of OsPXG9. Therefore, the observed relationships between OsPXG9 and antioxidant enzymes should be interpreted as correlations rather than definitive causations.

## 4. Materials and Methods

### 4.1. Vector Construction

The pGEX-4T1/OsPXG9 for recombinant protein expression and purification was constructed previously [19]. Briefly, OsPXG9 (LOC_Os06g14370) was amplified by PCR from leaves of 2-week-old rice seedlings and cloned into the commercial pGEX-4T1 vector (Cat. #GE28-9545-49, Cytiva; Appendix A and Appendix A).

For the generation of OsPXG9 overexpression (ox) lines, cDNA of OsPXG9 was amplified from pET-28b/OsPXG9 [19] by PCR, followed by digestion with NcoI and BstEII. The insert was then ligated to replace the GUS sequence in the commercial pCAMBIA1201 vector (Cat. #VET1345, Creative Biogene, Shirley, NY, USA) to make the pCAMBIA1201/OsPXG9-ox plasmid (Appendix A and Appendix A).

For generation of OsPXG9 knockout (ko) lines by CRISPR/Cas9, two sgRNAs (sgRNA-9 and sgRNA-41) were designed by CRISPR-P 2.0 (http://crispr.hzau.edu.cn/CRISPR2/ (accessed on 25 August 2020)) based on the genomic sequence of OsPXG9 (LOC_Os06g14370; Appendix A). The construction protocol was adapted from a previous publication [65]. Briefly, the two sgRNAs were separately constructed into the pRGE31 vector following the manufacturer’s guidelines, as follows: First, two single-stranded 70 nt DNA oligonucleotides (ssDNA oligo) containing sgRNA sequences were synthesized and purified to the final concentration of 0.2 µM, and the pRGE31 backbone vector was digested with BsaI (Cat. #R3773, NEB, Ipswich, MA, USA) for 16 h at 37 °C. Second, 30 ng of restriction enzyme-linearized vector and 0.2 ng of ssDNA oligo were mixed with 10 µL of NEBuilder HiFi DNA Assembly Master Mix (Cat. #E2621, NEB) with a final volume of 20 µL, and the mixture was incubated at 50 °C for 1 h. Next, the complex of U3 promoter, sgRNA-41, and gRNA scaffold was amplified by PCR and inserted into the pRGE31/sgRNA-9 vector to make the pRGE31/dual-sgRNA plasmid (Appendix A). Finally, to construct binary vectors for Agrobacterium-mediated rice transformation, the sgRNA cassettes (from pRGE31/sgRNA-9, pRGE31/sg-41, and pRGE31/dual-sgRNA) and pCAMBIA-Cas9 binary vector [65] were double-digested with HindIII (Cat. #R3104, NEB) and BglII (Cat. #R0144S, NEB) overnight at 37 °C, separated by gel electrophoresis, extracted, and then ligated with the ratio of vector and insert of 2:3 using Instant Sticky-end Ligase Master Mix (Cat. #M0370S, NEB) overnight at 4 °C. Finally, the ligated binary vectors (pCAMBIA-Cas9-sgRNA-9, pCAMBIA-Cas9-sgRNA-41, and pCAMBIA-Cas9-dual-sgRNA) were confirmed by colony PCR and DNA sequencing (Appendix A).

### 4.2. Heterologous Expression and Purification of OsPXG9

For expression and purification of OsPXG9, *E. coli* BL21 (DE3) codon plus cells were transformed with the pGEX-4T1/OsPXG9 vector and grown at 37 °C until the optical density (OD) reached 0.6~0.8 at 600 nm. Protein expression was induced with 1 mM isopropyl-ꞵ-D-1-thiogalactopyranoside (IPTG) for 16 h at 25 °C. The cells were harvested by centrifugation (5000× *g*, 15 min, 4 °C), resuspended in phosphate-buffered saline (PBS, pH 7.3) containing 0.6 mM PMSF and 0.2% Tween 20, disrupted by sonication, and centrifuged (15,000× *g*, 90 min, 4 °C). The supernatant was loaded on a glutathione Sepharose column (Cat. #17-0756-01, GE Healthcare, Chicago, IL, USA) and washed with 1× PBS. OsPXG9 was cleaved and eluted with PBS (pH 7.3, containing 0.2% Triton X-100) containing thrombin protease (100 units, Cat. #27-0846-01, GE Healthcare) according to the manufacturer’s instructions. Purified OsPXG9 was concentrated by centricon (Vivaspin 6 Centrifugal Concentrator, MWCO 10, Sartorius, Epsom, UK) and stored in 50 mM sodium phosphate buffer, pH 7.3, until use. The concentration of OsPXG9 was measured using a BCA assay kit (Cat. #23227, Thermo Fisher Scientific, Waltham, MA, USA).

### 4.3. Substrate Preparation, Enzyme Assay, and Kinetic Analysis

Hydroxy fatty acids (9-hydroxyoctadecadienoic acid (9(*S*)-HODE), 13-hydroxyoctadecadienoic acid (13(*S*)-HODE), 9-hydroxyoctadecatrienoic acid (9(*S*)-HOTE), and 13-hydroxyoctadecatrienoic acid (13(*S*)-HOTE)) were synthesized from linoleic acid (LA) or linolenic acid (LnA) as follows: First, 0.5 mM of L(n)A was converted to 9(*S*)- or 13(*S*)-hydroperoxyoctadecadi(tri)enoic acid (HPOD(T)E) by 9-LOX (CaLOX1, FJ377808) [33] or 13-LOX enzyme (Cat. #L7395, Sigma-Aldrich, Saint Louis, MO, USA) in 25 mL of 50 mM sodium phosphate buffer (pH 7.3), incubated overnight. The enzymatic reaction was stopped by adjusting pH to 3.0 with 1 M HCl, and the 9/13(*S*)-HPOD(T)E were extracted by liquid–liquid extraction using 25 mL of methylene chloride (4 times). Extracts were combined, and methylene chloride was evaporated using a rotary evaporator (Eyela, Tokyo, Japan). The 9/13(*S*)-HPOD(T)E were then reduced by an excess amount of triphenylphosphine (Cat. #T84409, Sigma-Aldrich) in 3 mL of methanol. The reaction was stopped by evaporating methanol using a rotary evaporator, and the triphenylphosphine oxide was precipitated by dissolving the reaction in 3 mL of hexane and removed by filtering through a filter paper. Hexane was then evaporated, and the hydroxy fatty acids (9/13(*S*)-HOD(T)E) were dissolved in methanol and stored at −20 °C until use.

For the enzyme assay, the consumption of conjugated diene in 9/13(*S*)-HOD(T)E by OsPXG9 was scanned by a UV spectrometer (UV02550, Shimadzu, Tokyo, Japan) at 234 nm. Hydroxy fatty acid and HOOH or CuOOH were added to 2 mL of sodium phosphate buffer 50 mM, at pH 7.3. Then, the reaction was started by adding 1 μg of boiled or active OsPXG9 (Appendix A).

To measure the kinetic parameters of OsPXG9 toward the oxygen acceptor (9/13(*S*)-HODE and 9(*S*)-HOTE), the enzyme reactions were performed in 2.5 mL of sodium phosphate buffer 50 mM, pH 7.3, with the presence of 1 mM hydrogen peroxide (HOOH) or cumene hydroperoxide (CuOOH) as the oxygen donor. For the kinetic parameters of OsPXG9 toward the oxygen donors, the reactions were performed in 2.5 mL of sodium phosphate buffer 50 mM, pH 7.3, with the presence of 20 μM hydroxy fatty acid (9/13(*S*)-HODE or 9(*S*)-HOTE) as the oxygen acceptor. To start the reaction, 1 μg of OsPXG9 was added to the mixture, and the consumptions of hydroxy fatty acids were recorded at 234 nm. The concentrations of 9/13(*S*)-HODE and 9(*S*)-HOTE were calculated using a molecular extinction coefficient of 25,000 (cm^−1^M^−1^) [34]. Kinetic constants (K_m_ and k_cat_) of OsPXG9 were calculated by the Lineweaver–Burk plot (Appendix A).

### 4.4. Extraction and Thin-Layer Chromatography (TLC) of OsPXG9 Reaction Mixture

An amount of about 10 μg of previously prepared hydroxy fatty acid substrates (9/13(*S*)-HPOD(T)) or unsaturated fatty acid (oleic acid (OA), C18:1; linoleic acid (LA), C18:2; linolenic acid (LnA), C18:3) and about 1 mM final concentration of HOOH or CuOOH were added to the total volume of 25 mL of sodium phosphate buffer 50 mM, pH 7.3, together. Boiled or active OsPXG9 (5 μg) was added, and the reaction was left shaking overnight at room temperature. The enzymatic reaction was stopped by adjusting pH to 3.0 with 1 M HCl and extracted with methylene chloride (25 mL) thrice. Extracts were combined and the solvent was evaporated. The extracted enzymatic reaction products were analyzed by TLC on a silica plate (Cat. #Z122785, Sigma-Aldrich). The TLC plate was developed with a mixture of hexane:ethyl acetate (3:7 (*v/v*) for hydroxy fatty acid reactions or 6:4 (*v/v*) for unsaturated fatty acid reactions) and stained with 10% phosphomolybdic acid.

### 4.5. Liquid Chromatography–Tandem Mass Spectrometry (LC-MS/MS) and Nuclear Magnetic Resonance Spectroscopy (NMR)

After extraction, the product mixture was dissolved in acetonitrile and analyzed by LC-MS/MS (High-Performance Liquid Chromatograph, Triple Quadrupole Mass Spectrometer, LCMS-8050 system, Shimadzu, Kyoto, Japan) using a C18 column (150 × 2.1 mm, 2.6 μm). The injection volume was 10 µL. The mobile phase consisted of solvent A (5 mM ammonium acetate and 0.1% formic acid in distilled water (*v/v*)) and solvent B (5 mM ammonium acetate and 0.1% formic acid in methyl alcohol (*v/v*)). The solvent gradient mixer was programmed as 0–1 min, 5% B; 1–3 min, 5–60% B; 3–8 min, 60–100% B; 8–11 min, 100% B, with a 0.3 mL/min flow rate, and oven temperature was 40 °C. Mass spectrometry was operated in ESI negative mode with the following parameters: detector voltage 1.84 kV for negative modes, interface voltage 4.0 kV, interface temperature 150 °C, desolvation temperature 260 °C, nebulizing gas flow 3.0 L/min, and heating gas flow 10.0 L/min. The mass data were collected in the m/z range of 100–500 for 11 min. MS/MS analysis was performed using collision energy with 25 eV to the selected mass ions. Total ion current with m/z range between 260 and 350 was swept by scanning mode to screen for reaction products. LC-MS data files (LCD format files), including MS and MS2 spectral data, were converted to mzXML files using MS Convert in the Proteowizard software Version 3.0.18205 [66]. The converted raw data were analyzed using MZmine version 2.53. For NMR analysis, the product mixture from the enzymatic reaction of 9(*S*)-HOTE and HOOH was loaded through a self-prepared silica chromatography column from silica gel 60, 230–400 mesh (Cat. #3912, Duksan, Seoul, Korea), and then was separated by a gradient ratio of hexane:ethyl acetate (9:1 to 1:9, *v/v*). The purified peak 8 was then subjected to analysis by Proton NMR with 400 MHz Bruker–NMR, and spectral data were analyzed by MestreNova 5.1.1.

### 4.6. Agrobacterium-Mediated Transformation

The japonica cultivar variety Ilmi (*Oryza sativa* L. Japonica cv. ‘Ilmi’) was used as the wild-type (WT) material for *Agrobacterium tumefaciens*-mediated transformation to generate the OsPXG9 overexpression (ox) lines and OsPXG9 knockout mutants by the CRISPR/Cas9 system. The constructed pCAMBIA1201/OsPXG9-ox, pCAMBIA-Cas9-sgRNA-9, pCAMBIA-Cas9-sgRNA-41, and pCAMBIA-Cas9-dual-sgRNA were transformed into the *Agrobacterium tumefaciens* (EHA105 strain) and co-cultured with the calli induced from mature seeds of rice. The selection and regeneration of transformed calli were carried out as previously described [67].

### 4.7. Plant Materials and Stress Treatments

Seeds from WT or OsPXG9-ox lines were germinated on Petri dishes with 5 mL of distilled water in the dark for 5 days. The germinated seeds were then transferred to the soil pots placed in a water reservoir. Plants were grown in a 16 h light/8 h dark cycle at room temperature. Three-week-old seedlings were subjected to stressors, as follows: For drought stress, seedlings were taken out of the growing pots and gently washed to remove soil. The treated plants were then kept in a dried sieve tray for 4 h (for DAB staining, NBT staining, and MDA content quantification) or 6 h (for Evan’s blue staining) at room temperature. For salt stress, 200 mM NaCl was added directly to the water reservoir containing the rice seedling pots for 24 h (for DAB staining, NBT staining, and MDA content quantification) or 36 h (Evan’s blue staining). All the treated rice seedlings were re-subjected to normal water condition for 12 h to recover.

### 4.8. Histochemical Staining and MDA Content Quantification

All the third leaves of WT and stress-treated seedlings were collected and directly subjected to freshly prepared and filtered 0.1% (*w/v*) nitroblue tetrazolium (NBT) solution (24 h), 0.2% 3,3′-diaminobenzidine (DAB) (*w/v*) solution (24 h), or 0.25% Evan’s blue (*w/v*) solution (30 min) at room temperature, followed by boiling in 95% ethanol for 4 h and observed under a light microscope with 10× magnification. Non-treated seedlings were used as the control. The stained area was converted to numerical values by ImageJ Version 1.45p software using three different microscopic photos of each sample.

For MDA content quantification, third leaves of the WT and stress-treated seedlings were collected, and MDA content was extracted and measured by the Malondialdehyde (MDA) Colorimetric Assay Kit (Cat. #E-BC-K025-S, Elabscience, Houston, TX, USA). Briefly, 15 mg fresh-weight leaf samples were ground thoroughly and extracted following the manufacturer’s guidelines. UV absorbance at 532 nm was measured by a UV spectrometer (UV02550, Shimadzu, Japan), and the MDA content was converted according to the guidelines.

### 4.9. RNA Extraction and Real-Time Quantitative Reverse Transcription PCR

For confirming the expression level of *OsPXG9* in *OsPXG9* ox plants, mature leaves from T_0_ and T_1_ putative mutant plants were collected. For checking the expression of oxidative stress-response-related genes, all the third leaves of the stress-treated seedlings were collected. All the samples were immediately frozen by liquid nitrogen and stored at −80 °C until use. Total RNA was extracted as described previously [65], and the cDNA library was synthesized from 1 µg of the extracted RNA using the QuantiTect Reverse transcription Kit (Cat. #205311, Qiagen, Venlo, The Netherlands), according to the manufacturer’s protocol. Each reaction contained 50 ng cDNA, 0.5 µM forward primer, 0.5 µM reverse primer, and 10 µL of 2× qPCR Master Mix (QuantiTect SYBR Green PCR Kit, Cat. #204343, Qiagen) in a total volume of 20 µL. Quantitative RT-PCR was performed on a Qiagen Rotor-Gene Q cycler, as follows: initial denaturation at 95 °C for 10 min, 40 PCR cycles at 95 °C for 30 s, 58 °C for 30 s, 72 °C for 30 s, and a melting curve analysis from 72 to 95 °C. Reactions were performed in triplicate. The expressions of transcripts of target genes were determined, and the expression signal was normalized to the expression of *ubiquitin 10* (*LOC_Os02g06640*). Data were expressed as a relative expression between treated and untreated samples (for checking the expression level of oxidative stress response genes) or between WT and *OsPXG9* ox plants (for confirming the overexpression of *OsPXG9*).

For determining the copy number, gDNA from the mature leaves of WT and two T0 transgenic plants was extracted and diluted to a serial concentration of 4 ng, 2 ng, 500 pg, 100 pg, and 20 pg. The qRT-PCR reaction was performed using gDNA with a similar composition and reaction cycle as described above. The copy number of OsPXG9 in WT and overexpression lines was determined according to the previously described method [68] using *sucrose phosphate synthase* (*SPS, LOC_Os01g69030*) as the reference for one copy in rice. All the primers are shown in Appendix A.

## 5. Conclusions

In conclusion, OsPXG9 is a Class I peroxygenase whose primary biological role is to reduce hydroperoxide species without consuming unsaturated free fatty acids. Importantly, transgenic rice plants engineered to overexpress OsPXG9 demonstrated higher resistance to drought and salt stress than WT control plants, and more effectively scavenged and reduced abundance of ROS after exposure to salt or drought stress. OsPXG9 also negatively regulated expression of other antioxidant enzymes by regulating the cellular redox state. Therefore, after exposure to oxidative stress, activated OsPXG9 displayed improved capacity to efficiently reduce lipid peroxidation, suppress production of ROS, and prevent apoptotic cell death, thereby improving the overall resistance of rice plants to drought and salt stress.

## Figures and Tables

**Figure 1 ijms-26-06918-f001:**
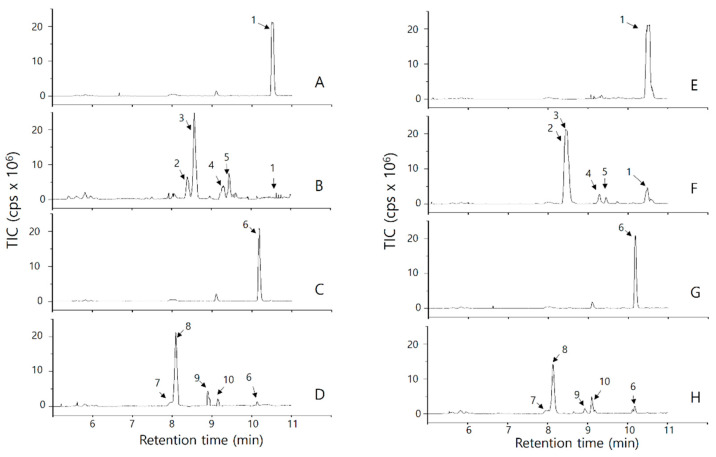
LC-MS/MS chromatogram of products of OsPXG9 catalysis on the 9-PXG pathway with HOOH (**A**–**D**) or CuOOH (**E**–**H**). The reactions by boiled OsPXG9 were used for the analysis of the substrates (9(*S*)-HODE (**A**,**E**) or 9(*S*)-HOTE (**C**,**G**)). The reactions by active OsPXG9 were used for the analysis of the enzymatic products from 9(*S*)-HODE (**B**,**F**) or 9(*S*)-HOTE (**D**,**H**). All the peaks indicated by the arrow and peak number were identified based on metabolome databases, and structures were confirmed by MS/MS data (e.g., fragmentation patterns (Appendix A)).

**Figure 2 ijms-26-06918-f002:**
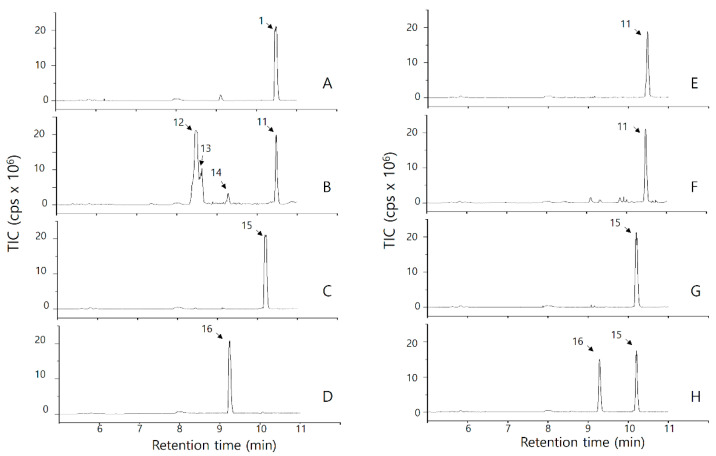
LC-MS/MS chromatogram of products of OsPXG9 catalysis on the 13-PXG pathway with HOOH (**A**–**D**) or CuOOH (**E**–**H**). The reactions by boiled OsPXG9 were used for the analysis of the substrates (13(*S*)-HODE (**A**–**E**) or 13(*S*)-HOTE (**C**–**G**)). The reactions by active OsPXG9 were used for the analysis of the enzymatic products from 13(*S*)-HODE (**B**,**F**) or 13(*S*)-HOTE (**D**,**H**). All the peaks indicated by the arrow and peak number were identified based on metabolome databases, and structures were confirmed by MS/MS data (e.g., fragmentation patterns (Appendix A)).

**Figure 3 ijms-26-06918-f003:**
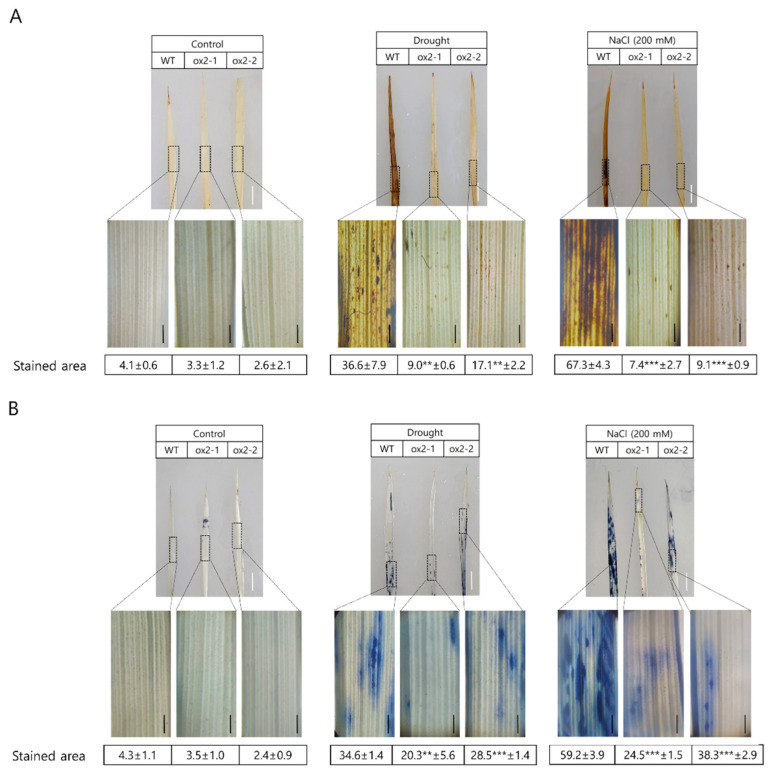
Accumulation of ROS and cell death level in rice seedlings from WT and ox lines under drought and salt stress. (**A**) DAB staining, (**B**) NBT staining, and (**C**) Evan’s blue staining. Three-week-old seedlings grown in soil were subjected to stressors, as follows: For drought stress, seedlings were taken out and dried in the air at room temperature for 4 h (DAB and NBT staining) or 6 h (Evan’s blue staining); for salt stress, 200 mM NaCl was added directly to the water reservoir containing the rice seedling pots for 24 h (DAB and NBT staining) or 36 h (Evan’s blue staining). All the treated rice seedlings were re-subjected to the normal water condition for 12 h to recover before staining with 0.2% DAB (*w/v*) solution (24 h), 0.1% (*w/v*) NBT solution (24 h), or 0.25% Evan’s blue (*w/v*) solution (30 min), followed by boiling in 95% ethanol for 4 h and observed under a light microscope with 10× magnification. Non-treated seedlings were used as the control. White bar = 1 cm and black bar = 1 mm. The stained area was converted from three separated microscopic photos using ImageJ Version 1.45p. (**D**) Malondialdehyde (MDA) content in the leaves of WT, ox2-1, and ox2-2 under drought and salt stress. Error bars denote ± SD of three replicates. The *p*-values were calculated using the Student’s *t*-test (** *p* < 0.05 and *** *p* < 0.01).

**Figure 4 ijms-26-06918-f004:**
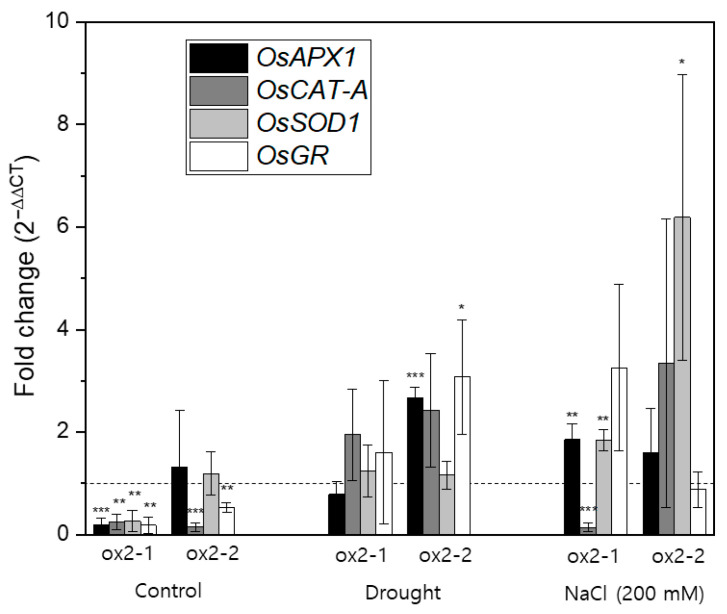
Gene expression patterns of antioxidant-related enzymes in OsPXG9 ox lines under drought and salt stress. Quantitative real-time polymerase chain reaction (qRT-PCR) assessed transcript levels of antioxidant-related enzymes, including ascorbate peroxidase (APX), catalase (CAT), superoxide dismutase (SOD), and glutathione reductase (GR), in the third leaves of the three-week-old rice seedlings after treatment. Drought and salt stress (200 mM NaCl) were treated as described in the Section 4. Materials and Methods Section. Non-treated seedlings were used as the control. The experiment was triplicated with ubiquitin (Os02g0161900) as the reference gene and wild-type (WT) as the reference sample. Expressions of the aforementioned genes in WT in control and stress conditions were used for normalization and presented as the dashed lines at 1. Error bars denote ± SD of three replicates. The *p*-values were calculated using the Student’s *t*-test (* *p* < 0.1, ** *p* < 0.05, and *** *p* < 0.01).

**Table 1 ijms-26-06918-t001:** Kinetic parameters of OsPXG9 ^a^.

Substrate ^b^	Oxygen Acceptor	9(*S*)-HODE					9(*S*)-HOTE				13(*S*)-HODE	
	Oxygen Donor	HOOH		CuOOH			HOOH	CuOOH			HOOH	CuOOH
k_cat_ (s^−1^)		20.1±	1.6	175.8±	11.1	8.3±	1.6	297.6±	122.1	96.1±	30.2	N/R ^c^
		5.8±	1.2	24.7±	5.1	7.8±	0.6	25.5±	2.7	0.6±	0.2	N/R
K_m_ (μM)		23.9±	4.8	84.8±	16.9	7.4±	1.8	231.4±	95.0	224.5±	35.0	N/R
		131.6±	56.7	99.2±	38.9	224.6±	27.8	57.6±	15.9	269.1±	128.6	N/R
k_cat_/K_m_ (s^−1^μM^−1^)		0.9±	0.1	2.1±	0.3	1.1±	0.3	1.3±	0.1	0.4±	0.1	N/R
		46.5 × 10^−3^ ± 9.6 × 10^−3^		260.8 × 10^−3^ ± 50.3 × 10^−3^			34.7 × 10^−3^ ± 2.6 × 10^−3^	454.9 × 10^−3^ ± 69.7 × 10^−3^		2.4 × 10^−3^ ± 0.4 × 10^−3^		N/R

^a^ Triplicates of data were used in the Lineweaver–Burk plot to determine kinetic parameters (K_m_ and k_cat_). ^b^ The consumption of substrates was monitored as follows: First, hydroperoxy fatty acids (9/13-HPOD(T)E) were generated from LA or LnA by 9-LOX (CaLOX1, FJ377808) [33] or 13-LOX (Cat. #L7395, Sigma-Aldrich), Saint Louis, MO, USA). Second, hydroxy fatty acids were produced from the corresponding hydroperoxide fatty acids by incubation in methanol and excess triphenyl phosphine (Cat. #T84409, Sigma-Aldrich). All reactions were quantified by measuring change in A_234 nm_. The concentration of fatty acid derivatives was calculated using a molecular extinction coefficient of 25,000 cm^−1^M^−1^ [34]. The catalysis of 13-HPOTE, 13-HOTE, and 13-HOTE by OsPXG9 did not break the conjugated diene of the fatty acid substrates, thus the kinetic parameters of those reactions could not be measured by this method. ^c^ N/R: no reaction.

**Table 2 ijms-26-06918-t002:** LC-MS/MS analysis of the products from hydroxy fatty acids’ metabolism by OsPXG9.

Peak Number	Retention Time (min)	[M-H]^−^ m/z	MS/MS ^a^	Chemical Formula	Molecular Weight	Mass Bank Score	Identification
1	10.52	295.2	[M-H]^−^: 295; [M-H_2_O-H]^−^: 277; [M-CH_3_(CH_2_)_4_(CH)_4_-H]^−^: 171	C_18_H_32_O_3_	296.2351	0.9555	9(*S*)-hydroxy-10,12-octadecadienoic acid (9(*S*)-HODE)
2	8.40	329.3	[M-H]^−^: 329; [M-H_2_O-H]^−^: 311; [M-H_2_O-H_2_O-H]^−^: 293; [M-CH_3_(CH_2_)_4_CH-OH]^−^: 229; [M-CH_3_(CH_2_)_4_CH-OH-H_2_O]^−^: 211; [M-CH_3_(CH_2_)_4_(CH)_3_-(OH)_2_]^−^: 183; [M-CH_3_(CH_2_)_4_(CH)_4_-(OH)_2_]^−^: 171	C_18_H_34_O_5_	330.2406	0.4548	9(*S*)-9,12,13-trihydroxy-10-octadecenoic acid (9(*S*)-9,12,13-THOE)
3	8.57					0.4806	
4	9.33	311.2	[M-H]^−^: 311; [M-H_2_O-H]^−^: 293; [M-CO_2_-H_2_O-H]^−^: 249; [M-CH_3_(CH_2_)_4_(CH)_2_]^−^: 211; [M-CH_3_(CH_2_)_4_(CH)_2_-H_2_O]^−^: 193; [M-CH_3_(CH_2_)_4_(CH)_3_-O]^−^: 185; [M-CH_3_(CH_2_)_4_(CH)_4_-O]^−^: 171	C_18_H_32_O_4_	312.2301	N/A	9(*S*)-10,11-epoxy-9-hydroxy-12-octadecenoic acid (9(*S*)-10,11-EHOE)
5	9.50	311.2	[M-H]^−^: 311; [M-H_2_O-H]^−^: 293; [M-CO_2_-H_2_O-H]^−^: 249; [M-CH_3_(CH_2_)_4_CH-O]^−^: 211; [M-CH_3_(CH_2_)_4_CH-O-H_2_O]^−^: 193; [M-CH_3_(CH_2_)_4_(CH)_4_-O]^−^: 171	C_18_H_32_O_4_	312.2301	N/A	9(*S*)-12,13-epoxy-9-hydroxy-10-octadecenoic acid (9(*S*)-12,13-EHOE)
6	10.19	293.2	[M-H]^−^: 293; [M-H_2_O-H]^−^: 275; [M-CH_3_(CH_2_)_4_(CH)_4_-H_2_O-H]^−^: 171	C_18_H_30_O_3_	294.2195	0.9420	9(*S*)-hydroxy-10,12-octadecatrienoic acid (9(*S*)-HOTE)
7	8.13	327.2	[M-H]^−^: 327; [M-H_2_O-H]^−^: 309; [M-H_2_O-H_2_O-H]^−^: 291; [M-CH_3_(CH_2_)_4_CH-OH]^−^: 229; [M-CH_3_(CH_2_)_4_CH-OH-H_2_O]^−^: 211; [M-CH_3_(CH_2_)_4_(CH)_3_-(OH)_2_]^−^: 171; [M-CH_3_(CH_2_)_4_(CH)_4_-(OH)_2_]^−^: 171	C_18_H_32_O_5_	328.2250	0.4540	9(*S*)-9,12,13-trihydroxy-10,15-octadecadienoic acid (9(*S*)-9,12,13-THODE)
8	8.18					0.4487	
9	8.94	309.2	[M-H]^−^: 309; [M-H_2_O-H]^−^: 291; [M-CO_2_-H_2_O-H]^−^: 247; [M-CH_3_(CH_2_)_4_-(CH)_2_]^−^: 211; [M-CH_3_(CH_2_)_4_-(CH)_2_]-H_2_O]^-^: 193; [M-CH_3_(CH_2_)_4_(CH)_3_-O]^−^: 185; [M-CH_3_(CH_2_)_4_(CH)_4_-O]^−^: 171	C_18_H_30_O_4_	310.2101	N/A	9(*S*)-10,11-epoxy-9-hydroxy-12-octadecadienoic acid (9(*S*)-10,11-EHODE)
10	9.18	309.2	[M-H]^−^: 309; [M-H_2_O-H]^−^: 291; [M-CO_2_-H_2_O-H]^−^: 247; [M-CH_3_(CH_2_)_4_CH-O]^−^: 211; [M-CH_3_(CH_2_)_4_CH-O-H_2_O]-: 193; [M-CH_3_(CH_2_)_4_(CH)_4_-O]^−^: 171	C_18_H_30_O_4_	310.2101	0.5902	9(*S*)-12,13-epoxy-9-hydroxy-10-octadecadienoic acid (9(*S*)-12,13-EHODE)
11	10.49	295.2	[M-H]^−^: 295; [M-H_2_O-H]^−^: 277; [M-CH_3_(CH_2_)_4_CH-OH-H]^−^: 195,	C_18_H_32_O_3_	296.2351	0.8635	13(*S*)-hydroxy-9,11-octadecadienoic acid (13(*S*)-HODE)
12	8.49	329.2	[M-H]^−^: 329; [M-H_2_O-H]^−^: 311; [M-H_2_O-H_2_O-H]^−^: 293; [M-CH_3_(CH_2_)_4_CH-OH]^−^: 229; [M-CH_3_(CH_2_)_4_CH-OH-H_2_O]^−^: 211; [M-CH_3_(CH_2_)_4_(CH)_4_-(OH)_2_]^−^: 171; [M-CO_2_-(CH_2_)_7_-CH-OH-H_2_O]^−^: 139	C_18_H_34_O_5_	330.2406	N/A	13(*S*)-9,10,13-trihydroxy-11-octadecenoic acid (13(*S*)-9,10,13-THOE)
13	8.54					N/A	
14	9.35	311.2	[M-H]^−^: 311; [M-H_2_O-H]^−^: 293; [M-H_2_O- CO_2_-H]^−^: 249; [M-CH_3_(CH_2_)_4_CH-OH-H]^−^: 211; [M-CH_3_(CH_2_)_4_(CH)_3_-OH-H]^−^: 183; [M-CH_3_(CH_2_)_4_(CH)_3_-OH-CO_2_-H]^−^: 139; [M-CH_3_(CH_2_)_4_(CH)_4_-OH-H]^−^: 171;	C_18_H_32_O_4_	312.2301	N/A	13(*S*)-9,10-epoxy-13-hydroxy-11-octadecenoic acid (13(*S*)-9,10-EHOE)
15	10.20	293.2	[M-H]^−^: 293; [M-H_2_O-H]^−^: 275; [M-CH_3_(CH_2_)_2_(CH)_2_-H]^−^: 223; [M-CH_3_(CH_2_)_2_(CH)_3_-OH-H]^−^: 195	C_18_H_30_O_3_	294.2195	0.9417	13(*S*)-hydroxy-9,11,15-octadecatrienoic acid (13(*S*)-HOTE)
16	9.29	309.2	[M-H]^−^: 309; [M-H_2_O-H]^−^: 291; [M-CH_3_(CH_2_)_2_(CH)_2_-O]^−^: 223; [M-CH_3_(CH_2_)_2_(CH)_3_-OH]^−^: 195; [M-CH_3_(CH_2_)_2_(CH)_2_-O-CO_2_]^−^: 179	C_18_H_30_O_4_	310.4292	N/A	13(*S*)-15,16-epoxy-13-hydroxy-9,11-octadecadienoic acid (13(*S*)-15,16-EHODE)

^a^ MS/MS fragmentation patterns are shown in Appendix A. N/A: not available from the mass bank library (https://massbank.eu/ (accessed on 3 February 2025)).

**Table 3 ijms-26-06918-t003:** Relative abundance of reaction mixtures in the 9- and 13-PXG pathways catalyzed by OsPXG9.

Path	Oxygen Acceptor	Oxygen Donor	Product	K_rel_
9-PXG	9(*S*)-HODE	HOOH	9(*S*)-9,12,13-THOE	22.6
			9(*S*)-10,11-EHOE	4.0
			9(*S*)-12,13-EHOE	4.7
		CuOOH	9(*S*)-9,12,13-THOE	5.3
			9(*S*)-10,11-EHOE	0.4
			9(*S*)-12,13-EHOE	0.1
	9(*S*)-HOTE	HOOH	9(*S*)-9,12,13-THODE	12.7
			9(*S*)-10,11-EHODE	2.0
			9(*S*)-12,13-EHODE	0.9
		CuOOH	9(*S*)-9,12,13-THODE	7.1
			9(*S*)-10,11-EHODE	0.7
			9(*S*)-12,13-EHODE	1.8
13-PXG	13(*S*)-HODE	HOOH	13(*S*)-9,10,13-THOE	3.0
			13(*S*)-9,10-EHOE	0.2
	13(*S*)-HOTE	HOOH	13(*S*)-15,16-EHODE	31.7
		CuOOH	13(*S*)-15,16-EHODE	0.9

The equilibrium constants for the reactions were calculated from apparent peak areas in the LC chromatogram using the formula K_rel_ = [product]/[substrate].

## Data Availability

Data are contained within the article and Appendix A.

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
