# Peer review of "Rice Peroxygenase-9 Negatively Regulates Production of Reactive Oxygen Species and Increases Cellular Resistance to Abiotic Stress"

_ijms, 2025, doi:10.3390/ijms26146918_

Round 1

Reviewer 1 Report

Comments and Suggestions for Authors

Tran et al. reported both the in vitro and in vivo functional characterization of OsPXG9 in rice. Tran et al. combined enzyme kinetics, metabolite profiling, transgenic overexpression, and stress assays to support the hypothesis that OsPXG9 declines lipid peroxidation, represses ROS production,  and prevents apoptotic cell death, thereby improving drought and salt stress resistance in rice. This study is overall promising, but major revisions are required for data clarification and interpretation.

  1. Line 19-23: This sentence is too dense. Consider split into shorter sentences.
  2. Line 69-77: Consider clarify what new insights are added here compared to the prior work.
  3. Table 1: The kinetic parameter results are informative but wound benefit from clearer visuals. Consider provide a schematic summarizing substrate preferences, and discuss how these properties may relate to physiological relevance.
  4. Table 3: In the footnotes, clarify how Krel was calculated.
  5. Lines 217–236: Only two overexpression lines were analyzed, and no successful knockout lines were obtained. This limits the genetic evidence supporting functional conclusions. Emphasize this limitation explicitly in the Discussion (Lines 384–424) and consider reframe conclusions more cautiously as correlation rather than causation.
  6. Figure 4: The variability of antioxidant gene expression among ox2-1 and ox2-2 is intriguing but under-discussed. Expand the interpretation—are these changes likely compensatory, indirect effects, or evidence of functional crosstalk?
  7. Line 237-239: Quantitative data should accompany staining photos. Provide mean signal intensity ± SD and p-values.
  8. Line 244-248: State explicitly if differences were statistically significant (for example, p-values?) for each comparison.
  9. Line 275-276: Provide exact fold changes, and statistical significance.
  10. Line 296: Clarify why p<0.1 is used as threshold. Typically, * stands for p<0.05.

Author Response

Please see the attachment file "250711 Response to Reviewer 1"

Reviewer 2 Report

Comments and Suggestions for Authors

This paper examines the effects of rice peroxygenase-9 on the production of reactive oxygen species and plant tolerance to abiotic stresses such as drought and salinity. The authors show that transgenic rice plants engineered to overexpress OsPXG9 exhibit enhanced tolerance to drought and salt stress, and are more effective in reducing ROS after exposure to salt or drought stress. OsPXG9 also negatively regulates the expression of other antioxidant enzymes by regulating the redox state of cells, demonstrating an enhanced ability to effectively reduce lipid peroxidation, suppress ROS production, and prevent apoptotic cell death, thereby enhancing the overall tolerance of rice plants to drought and salt stress. The paper is clearly written, at a high scientific level, and describes both the methods used and the results obtained well. The review is fairly comprehensive. This work will contribute to the understanding of the specific molecular mechanisms that allow rice peroxygenase-9 to provide enhanced plant tolerance to abiotic stress. Unfortunately, I was unable to obtain supplementary materials for this article. There are no other major comments.
Below, specific minor comments are described.
1. Line 19. The abbreviation ROS needs to be defined in the text.
2. Line 87 et seq. Supplementary materials are missing.
3. Figure 3. The figure is too large; it would be better to break it into smaller ones.

Author Response

Please see the attachment file " 250711 Response to Reviewer 2"

Round 2

Reviewer 1 Report

Comments and Suggestions for Authors

This is the revised manuscript by Tran et al. In this version, the authors have addressed all of my concerns. I do not have additional comments.